# Did We Need Another *Emma*? The Anxiety of Influence in the Bollywood Adaptation of *Emma*

**Meenakshi Bharat**

Department of English, University of Delhi, Delhi 110007, India; mbharat@svc.ac.in

**Abstract:** The multiple screen adaptations of Jane Austen's novels, and in particular, those of *Emma* (1815–1816), willy-nilly direct audience attention to the problematic continuities between the original novel and Rajshri Ojha's twenty-first century Bollywood adaptation, *Aisha* (2010). This essay addresses the issue of the competing influence of Austen and the global cinematic adaptations that precede this Hindi adaptation, even as it assesses the film for its engagement with the adaptation of Austenian social concerns to the particularities of the contemporary upper-middle-class urban existence in India.

**Keywords:** Jane Austen; Indian cinema; Bollywood; *Emma*; *Aisha*; adaptation

One of the most vexing issues about engaging with an adaptation is the pressure to reckon with the general morality of the exercise, and to essay, what James O Young calls, a 'moral assessment' of the relationship of the particular adaptation with the original (Young 2005). However, if the adaptation exercise seems to imply 'a profound offence' in cultural appropriation from one perspective, from another, it can be seen as an organic, 'rhizomatic assemblage', to adapt Gilles Deleuze and Félix Guattari's critical formulation.[1] Three patently distinct artistic units—one, an early-19th-century novel, the second a late-20th-century Hollywood offering, and the third, a turn of the millennium, early-21st-century Bollywood engagement, with the preceding original and its Western adaptational version—tender a tantalizing invitation to wrestle with this conundrum. What is the thread connecting the English Jane Austen's *Emma* (1815)[2], the American Amy Heckerling's *Clueless* (1996), and the Indian Rajshri Ojha's *Aisha* (2010)? Are the followers, especially *Aisha*, '[un]suitably [in]discreet', as Young suggests adaptations sometimes can be, or do they display the organic 'multiplicity', marked by 'principles of connection and heterogeneity' in the Deleuze and Guattari mould? (See Note 1 above).

## 1. The Austen Lure in the Indian Subcontinent

Macaulay's Minute's push for the introduction of English as the medium of education in India in 1835 opened the sluice gates for the dissemination and popularization of English literature as part of the concerted literary-studies programme in the colony. Jane Austen, Britain's favourite early nineteenth-century woman novelist, was a cornerstone of this rich cultural gift hamper of English literary works. Notwithstanding the problematic intimations of a total cultural takeover of the colony, colonial subjects, exposed to an entirely fresh cultural perspective through the richness of British literature, took to her with alacrity. They embraced the British pride in and fascination with Austen's works, making it their own, and this allure percolated down through the centuries, its deep impact keeping in step with the shifting times. As in the mother country, acquiring an exceptional hue in technologically advanced times, the Austen novels have now come to be widely adapted both for television and the big screen in its former jewel in the crown in South Asia. Today, as Diane Sadoff remarks, 'Filmmakers incorporate and indigenize Austen's stories and heroines for national and diasporic South Asian spectators', as a result of which 'Austen

has now become a product brand [providing] pre-sold content for a converged, synergistic multimedia entertainment industry across global markets' (Sadoff 2010).

Over the years, in India, this unfaltering global, adaptive engagement with Austen has found remarkable continuity in extremely successful India-centric adaptations. Roundly shaking off the denigrating Woolfian appellation of cinema being a parasitic artform feeding on literature, its 'prey' and 'victim' (Woolf 1926) and Rabindranath Tagore's mounted attack against cinema 'still playing second fiddle to literature' and their call for the need to 'emancipate cinema from this bondage' (quoted in (Hutcheon 2006)), these several adaptations of Jane Austen's novels indicate that these apprehensions were quite unfounded. Rather, the steady subcontinental filmic attention on Austen's works is proof enough of the fact that each effort represents, to a greater or lesser extent, what Hutcheon names a 'process of creation [involving] both (re-)interpretation and (re)creation . . . [both] appropriation and salvaging'.[3]

While Austen's *Pride and Prejudice* (1813) has seen the highest number of screen portrayals and adaptations, *Emma* follows close behind with as many as nine global adaptations since 1948: six films and four television series.[4] Following the chartbusting history of the Hollywood *Clueless* (1995), the popularity of Austen as a subject for screen adaptations fuelled a wave of what has been variously, and quite disparagingly labelled as part of the 'Austenmania' sweeping across the world.[5] Disparaged by leading Indian film critics such as Kaveree Bamzai as an 'Austen-tatious' consumption of the literary original, *Aisha* has excited an equal number of negative reactions as applause (Bamzai 2010). Even as all the general reservations about the exercise of adaptation are called up, *Aisha* especially falls prey to the challenge of being measured up against not only all prior Austen adaptations but also more particularly against the adaptations of *Emma* that have preceded it. This sizing up is rather confounding and unsettling, raising the basic question of whether the Bollywood production, *Aisha*, should be seen as an adaptation of the original 1815 Austen literary offering or of the 1995 film *Clueless?* Should the Janeite supremacy of Austenian texts be the only guiding principle for assessing this Bollywood Austen adaptation, or should the recognition of *Clueless,* the Hollywood adaptation of the book and a modern cult classic, inform our reckoning of this Indian version? It is significant that the recent one-hour-long reunion of the *Clueless* cast organised by the Chicago Comic and Entertainment Expo, C2E2, in 2019, paying obeisance to its iconic status, seems to ride rough-shod over the literary ancestor by simply not paying it any attention. Can *Aisha* afford to ignore either one or the other?

## 2. India-Centric Austen Adaptations and Intriguing Case of *Aisha*

So, by the time *Emma* came to be adapted in Hindi cinema as *Aisha* (Rajshri Ojha), in 2010, there was already a well-established tradition of Austen screen adaptations, not only around the globe but also on the Indian subcontinent. Austen had already proved their eminent adaptability to Indian contexts by the string of notable multilingual, multicultural Indian screen adaptations of their novels for both television and the big screen. With an early promising beginning in 1985 in the popular *Trishna,* a serialised Hindi adaptation of *Pride and Prejudice* for Indian national television, this tradition found successful consolidation in two big-screen adaptations: one, of *Sense and Sensibility* as the Tamil *Kandukondain Kandukondain* (2000), and the second, of *Pride and Prejudice* as *Bride and Prejudice/Balle Balle Amritsar to L.A.*, the internationally acclaimed British–Indian adaptation in English and Hindi. *Aisha* is the most recent Austen adaptation, and Rajshri Ojha, its director, is presently working on another: *Sense and Sensibility*.[6] The double-edged implications of the development of this local Austen-adaptation tradition are all too apparent. Even as it provides a rock-solid confidence-boosting take-off point for fresh creativity, it throws it into a savage arena of unavoidable unforgiving comparisons—most often disparaging—not only with the original but also with all prior adaptations.

All screen adaptations of literary texts draw attention to the changes wrought in the journey from the linguistic sign in the word to the visual sign of the onscreen image,

whether through compression, omission, or extension. However, in the case of *Aisha*, the reinterpretation and reassessment process must necessarily also go back to include the visual text already constructed and popularised, since the resonances of the earlier filmic adaptation in *Clueless* are as strong, if not stronger, as those of the book. The mire of the confusions and obfuscations galore generated by the resultant complexities of the adaptational exercise, in which *Aisha* finds itself, demands that it be given a fresh look. Following the mutational trajectory through the three intermedial, multicultural, and multilingual versions straddling three centuries, intriguing questions regarding the implied silencing and enhancing are generated. What does the time lag between the initial publication of the book in 1815, its cult Hollywood adaptation in 1995, and the 2010 making of *Aisha* imply? What does it mean in terms of the subject matter and its artistic treatment? Evidently, the temporal shifts as well as the obvious geographical and cultural transference—where the adaptation was made, where the plot is located, and the choice of language for the adaptation—all have far-reaching consequences in the character of the end-product and its reception.

While it cannot be gainsaid that *Aisha* effectively taps the discovery of adaptation as a most fecund space for artistic creativity, it also points to a major limitation in its critical reception, in that it has mostly been seen only as an adaptation, as an update of Austen's original. Its identity as an independent creative entity has been largely ignored. The fraught tension inherent in being hemmed between being seen as an Austen 'upgrade', as looking back to the original and being recognised as an independent new work of art, informs an alert reading of the film.

A first necessary requirement for a fuller reading of *Aisha* is the recognition of the identity of the intended audience of this cinematic project. The broad Bollywood format signals that the movie is aimed at the typical Bollywood audience, familiar with the generic trappings of song and dance, among other things, of the Mumbai film. Moreover, very particularly and very obviously, it is aimed at a niche English-educated Indian-urban-elite Bollywood audience, which is also quite likely to be familiar with its Hollywood counterpart. This smart Hindi adaptation forces *this* audience to turn a critical eye on itself, by highlighting attributes that diminish it, even as it directs their attention to what is valuable. This is achieved admirably by the unique commitment of the filmic venture to adhere to an authentic representation of the immediate contemporary setting. So, it functions both as a gently subversive portrayal of the social class and the times that it is addressing—the dumb and girly pursuits of the twenty-first-century uber-rich—and as an eye-opener to the possibilities of their lot. The applause that Jane Hu gives to the *Clueless* Cher (Emma counterpart) and the adaptive world that she inhabits by quoting her quoting 'that Polonius guy' in *Hamlet*—"to thine own self be true", works admirably for Aisha (the Indian Emma) and her world.

Determinedly advertising the film as an adaptation of Austen right from the beginning, *Aisha* is the latest endorsement of Austen being exceptionally adaptable to the Indian scene and mainstream Bollywood. Even as the marketing ploy of harking back to Jane Austen in article after article and interview after interview, the many evident scenes lifted from *Clueless* and the intervening Hollywood precedent plunge both the maker of the film and the recipient, audience, and critic alike into what I see as an anxiety of influence. In addition, as often happens with cinematic adaptations, comparatist critique limited the immediate appreciation of its deeper adaptive achievement. On its release, Sukanya Verma, reviewing it for Rediff.com, declared, 'If one tries to dissociate Aisha from Austen and perceive it as a standalone rom-com, far less faults are to be found' (Verma n.d.). However, Ojha, far from being weighed down by the churn of what to be true to and what to break away from, endeavours and manages to turn the legacy to their advantage in more ways than one. One of the most obvious instances is the use of a protagonist voiceover, which is as perfectly in pitch as the narrative voice of Alicia Silverstone in *Clueless* had been, and which, in turn, is in conjunction with the self-aware commentary of Austen's Emma, thus marking

the continuity of the Austenian ironic slant and the self-turning critiquing of their leading 'heroines', Elizabeth Bennet and Emma Woodhouse, into the adaptations.

### 3. Living in a Bubble: Class Difference in *Aisha*

It is salutary to note that critical attention from the Western Hemisphere on *Aisha* has come up with various unique and interesting insights. However, in addition to being already plagued by the prevailing constraining pressures of reading adaptations, the inability of the Western critic to completely enter the world of upper-class Delhi resident, Aisha, majorly hampers an accurate reading. A purist Austen outlook has further exacerbated the problem, since it closes the door to deviations from the original. The compounding roadblocks to a full understanding especially cut off 'alien' adaptational readings. For me, this marks an open invitation to take its 'Indian' context into account to see what new light it throws on the original. What does *Aisha* accomplish, what does it do different, or what does it carry over from earlier versions that makes it a reckonable artistic entity? Or, to put it succinctly, does this 'version' of *Emma* accomplish anything at all? Or is it only a Bollywood dumbing down of the precursors?

Even as *Aisha* carries over the themes of its artistic predecessors, the film is significant for openly piggybacking on Austen and the Austen franchise. It also bears cinematic witness to the fact that by the turn of the century, urban, educated Indians have become articulate, confident, and comfortable in their own space in a markedly liberal world economic scenario. Here, buying and sporting international luxury brands, such as Louis Vuitton, Dior, and Jimmy Choo have now become a way of upper-middle-class existence, in much the same way as they have in other affluent societies across the globe. Moreover, as already remarked, mounted mainly for an urban viewership and representing their values in a new, emergent India, this adaptation, with its decided upper-class bearing, shuns mean, 'limited' country values and manners as represented by the gauche Shefali, who calls 'paintings' by the rather gawky appellation of 'drawings' and naively reveals how she has come to town only to net an eligible match. This gives a fresh turn, even though inverted, to the country/suburban/metropolis thrust of the *Emma* Highbury–London connection.

To start with, while *Aisha* picks up the materialism and class divides of Austen's world, the Indian setting gives the themes a peculiarly Delhi–Indian spin. *Aisha* effectively broaches the issue of class by identifying finer class distinctions, not discernible to people unfamiliar with the scene. The lumping together of Aisha–Arjun (Emma–Knightley) with Randhir (Mr. Elton) as members of an 'embarrassingly nouveau riche' social circle, by critics such as Theresa Kenney, is a case in point (Kenney 2011). To an Indian, especially to a Delhiite, the dynamics of in-class differences are all too clear. Randhir, the Mr. Elton equivalent, may be wealthy, but his 'sweetmeat business' puts him at a level below the established old-rich category to which the Kapoors (Woodhouses) and the Burmans (Knightleys) clearly belong. It is true that, fortified by the confidence that his wealth gives, Randhir never seems apologetic for his comparatively mean background, attending the parties and weddings hosted by the 'uber' class—Golf Links–Jor Bagh types, who go for jogs in heritage surroundings, as do Arjun and Aisha. While the old rich are much akin to the landed gentry, the Woodhouses of *Emma,* Randhir is clearly not their neighbour, not of their ilk. His devotion to Aisha is sincere, but that does not change the fact that Aisha belongs to a class that is a notch above his own. The fact that they have long known each other is witness to shifting class distinctions in modern Delhi society, most likely a result of schools bringing them together. On the other hand, Arjun and Aisha have been long-time neighbours, walking in and out of each other's homes with ease and comfort. The marriage between Aisha's sister, Aaliya, and Arjun's elder brother has cemented the bond. However, the same facility is not given onscreen to Randhir. He may have managed induction into the polo-playing Delhi royalty by dint of his material well-being, but it is not his natural social home, making him the one prototypical nouveau-riche specimen in the film. He appears socially awkward—his horse throws him off and he is left looking sheepish—whereas Arjun is in his element, the slick handsome goal-maker, who wows the

audience. Similarly, the country-cousin gaucherie identifies Shefali (Harriet) as a member of a different class altogether, and in the unthinking narcissistic world-view of the rich, such as Aisha, who think of themselves as the evolved norm, Shefali needs social education to elevate her.

Language is another identifier of the new-found self-assurance of the successful modern Indian. In the postcolonial carryover of the former masters' language, characters are now seen to be using English confidently as their *lingua franca*. However, when, moulding the language to their own environment, the characters speak the popular Delhi variant of 'Hinglish', which is no longer an apologetic lamentable spillover from the colony but is rather a confident, self-assured use of a hybrid version of the language. Interestingly, it is the subtle variations in this Hinglish that mark the class distinctions, very ably mirrored in Bhagat's dialogue. One major class signifier is the inflectional changes in language attributed to each character. Randhir speaks a more decidedly 'Punjabi' Hinglish, as compared to the smoother flow of the one spoken by the Kapoors and the Burmans. It is he who makes the nouveau-riche assertion of 'Western wear' being 'the dress code' at the polo match to Shefali (in a desperate bid to thereby imply social distance between himself and her), but when he sees Aisha and Pinky exchange glances, he backs off. The Shefali–Harriet character speaks a Haryanvi variant, labelled 'behenji' by Aisha and Pinky—her hick, lowly, and less well-endowed 'mofussil' class being, thus, repeatedly called up dialogically right from the beginning. The novel is less clear in marking the Harriet–Emma difference of class in the language they speak, though the *Clueless* Tai did voice country-cousin concerns. However, the Aisha variants are remarkable in registering this difference in class.

Accordingly, the alert dialogue also keeps harping on the material differences between the three girls: the recurrent appellation of Shefali as 'bechari', meaning 'poor', by Aisha, wraps in both her lower economic and social status. Aisha needs to use her add-on credit card to foot the bill for the modern, chic, designer clothes necessary to effect a transformation of Shefali. At the end of the film, this is quoted by Shefali herself as signifying the class difference that causes a rift between the two. When Shefali confesses that she is in love with Arjun, euphemistically alluding to the class differences, Aisha jealously tries to put her off, 'You and Arjun are very *different*. He is not your *type*'. It is then that, for the first time, Shefali turns around to roundly accuse Aisha of classist beliefs, of thinking that Arjun is not in her league simply because 'her father does not have the same kind of money' and because she belongs to the 'middle class'. She even accuses Aisha of thinking that she is a 'ganwaar', a country bumpkin, who needed to be taught social niceties. In a late epiphanic realization, she hurls an accusation at Aisha:

> I understand everything Aisha . . . Enough is enough, Aisha. You have never ever thought of me as your equal, isn't it?

It emerges that her true equal, her 'type', is Saurabh, the boy who has always loved her and for whom she too had always nurtured a soft spot, who also comes from the same social stratum. This is in line with *Emma*'s emphasis on a successful marriage of equals in ability, as represented by the Emma Woodhouse–Knightley and the Elizabeth Bennet–Darcy pairing.

The dialogically emphasised social origins and locations of the characters is backed by the screenplay, further highlighting the difference by the act of 'omission'. We are never taken to the shop of Randhir, the 'halwai' (sweetmeat vendor), or introduced to his home setting. Neither do we ever see Shefali's home. While Arjun is a regular visitor to Aisha's home (her father calls him 'ghar ka baccha', 'a child of the family'), raiding the refrigerator, sitting down to meals, or even watching the television, Randhir is never allowed this facility. Shefali may have been allowed entry to Aisha's home and room, but it is only under sufferance, due to the 'project' of transformation that the latter has undertaken.

*Aisha* comes then, as a mirror to a class of society that lives in a bubble, the accusation that Arjun, knowing her better than anyone else, levels at Aisha, 'You live in your own world'. Aisha is limited within its confines, and not being able to think otherwise, inflicts unintended hurt. It is this lack of knowledge and direction that gives the drive to the social

imperative of the film within which the character delineation, social critique, and social objectives are embedded. Voiced most often clear-sightedly by Arjun Burman, one who is an insider, an inhabitant of this world, this is indicative that all is not reprehensible with this world and there are inherent positive possibilities of improvement. While his Wharton education and his high-profile investment-banking job, position him in the premier economic and business hub of the country and take him all over the world, bearing witness to the economic upward mobility of young India, his alertness to Aisha's shortcomings and his willingness to be upfront about pointing them out to her is a sure sign that the well-being of the nation is in good hands. The fact that he loves Aisha even as he is critical of her indicates that there is something sound even in her character.

## 4. Loving Aisha—The 'Aisha' Value System

In taking the name of Aisha for its title, the film dignifies the character and gives her voice the perspectival thrust of the film, and the sympathy, thus, gained, testifies to the possibility of betterment and the soundness of underlying social and moral principles. Labouring under the Elizabeth Bennet kind of pride and prejudice, she, like Lizzie, is yet ultimately willing to re-think and re-orient her outlook. She is not viciously malicious, and the hurt she causes is not only unintended but also perversely born out of her misplaced sense of doing good. Even when she seems to speak the sometimes-unfeeling language of youth, such as the Cher of *Clueless* ('as if', 'whatever'), and seems to be nothing but a spoilt brat, her heart is clean and in the right place.

The opening voiceover gambit of the film itself is indicative of the sound value-system of this society—a value-system to which Aisha wholly subscribes. She may be as much the spoiled modern urbanite as Cher Horowitz, but in inhabiting the circumscribed socio-geographical space of Delhi's upper-class society, she has much that makes her kin to Emma. Far from living in a twenty-first-century valueless social set-up, she harkens back to the traditional values that have kept society whole and healthy. The director Rajshree Ojha and screenwriter Devika Bhagat astutely plumb the depths of this close-knit world of maasis and maasus, of sisters-in-law and brothers-in-law, of extended families that embrace brothers of brothers-in-law and sons of 'maasus', and of childhood mates and new friends.[7] Her feelings for her 'maasi' (mother's sister, the word literally meaning 'like mother'), who has taken care of her ever since her mother's death, opens her heart to immediately embracing and welcoming the colonel to the familial fold as her 'maasu'. This maasi (Miss Taylor), a dependable life advisor to Aisha, is one adult in whom she can and does confide her innermost feelings: she tells her that she is in love with Arjun. In line with this, Aisha is shown to be especially close to her father, with whom she can similarly share her deepest feelings at a time when her heart is breaking. Through the stock-Bollywood exchange between an understanding, supportive parent and a young child in the midst of stupendous emotional upheaval, the film, by upholding family values, manages to make a positive identification of the healthy core of Indian society. However brash the modern Aisha may sound in her threat of not ever wanting to marry (an avowal that the original Emma makes too) and of making do with a 'live-in relationship', her matchmaking instincts indicate otherwise: she sets great store by the institution of marriage. The way the assertion is flung, in a fit of momentary pique at her father's mild reprimand regarding her excessive spending that may not leave any money 'to get her married', gives the lie to it, as a statement not to be taken seriously.

Even her 'do-gooding' instincts ratify this inherent goodness of heart. It is true that 'do-goodery' is a known fashionable pastime of the rich, desperately looking to fill the boredom of their privileged existence. However, it is equally true that not all rich people indulge in this pursuit. Consequently, there is no reason why this evident honesty of intention should be discounted. It is notable that there are enough examples of prototypical, self-centred female protagonists in 'Bombay' cinema, such as 'Poo', Pooja in *Kabhi Khushi Kabhi Gham*,[8] who display no such social drives. Following in their wake, Aisha's impulse to do good may seem flighty at times –sometimes rescuing animals and sometimes 'poor'

people such as Shefali—but her sincerity is never in question. The innate goodness in her character is sublimated in her final self-realization, which becomes the critique of this self-involved world. However 'shallow' Aisha may seem initially, she is still an exemplar of the core value system of 'responsibility' as well as connection to family and to friends that runs through her world. Significantly, it is her misplaced sense of social responsibility, the 'social work', that occasions Aisha's major education and the sphere in which the growth of her character lies, quite in sync with Cher's efforts to lead a purposeful life following the realization of her love for Josh. Successfully sidestepping the rut of the 'do-gooding' socialite stereotype, her desire to do good is actually the 'substance' in her character that takes her to her final realization.

Moreover, and this is most important, for all her ill-advised, frivolous, and privileged pursuits—her shopping sprees, her yoga, river rafting, and parties—pursuits of Delhi's rich, Aisha is never the vacuous socialite, or, to use the Cher term, the clueless 'airhead'. Though Kenney too recognises Aisha as 'an intelligent and, eventually, reflective heroine', Aisha's sensibility embraces much more than the awakening of her heart 'to its own love of virtue, loyalty, and care as manifested in the long-known and thus unrecognized hero' that Kenney limits her to.[9] Self-aware Aisha realises the wrong that she has wrought without intending it. In the film, and this is crucial, she says sorry to each and every person she has wronged: her childhood friend Pinky, Randhir, Shefali, and Arjun. She is not ashamed of saying 'I am such an idiot', and acknowledging with chagrin that she has been 'very selfish', 'very self-centred', 'arrogant', and 'absolutely wrong'.

Backing this evolution in Aisha's character is the move towards the notion of economic independence, a notion that has no place in *Emma*, since Emma is secure in her landed gentry class and a thought like that never enters Cher's head. Notably, even at the outset, despite the fact that she lives with her rich parent, using facilities provided by him, and being economically dependent on him (she apparently has an add-on, supplementary credit card, with its expenses billed to her father), Aisha is still able to think for herself, fully aware that 'money doesn't grow on trees'. She shows herself to be a free-thinking, independent-minded individual, however misguided she may be, and can apparently manage on her own and manoeuvre herself out of tricky positions. Interestingly, it is within this paradoxically aspirational lifestyle that she learns about its limitations, with her outlook changing from a happy-go-lucky taking-everything-for-granted attitude to one that is more responsible. Even early on, we learn that she is a painter of no mean talent, her work being publicly displayed in an art gallery. An unimpeachable alibi for her abilities comes from Arjun himself, the most sensible person onscreen, 'You have so much potential, Aisha. There's so much you can do with your life'. In pointing out that her pursuits are 'shallow', he implies that she is not, and that she can move to acts of deeper meaning. Early in the film, sick of her interfering with matchmaking, he tells her in exasperation, 'Go get a job, Aisha', obliquely indicating that she can. In addition, in the final sequences of the film, chastened and wiser, she does acquire one. She is packed and ready to take up a job, most significantly not in Delhi, her hometown, with all its support system, but in distant Mumbai, marking a major forward progress, mentally and economically. It is significant that with all three creators being women—Austen, Heckerling, and Ojha—an unintended genealogy is implied, but *Aisha* gives a millennial turn to Jane Austen's satirical interrogation of the gendered burden of a circumscribed, patriarchal society. There is an underscoring irony in the fact that despite being dependent on her father to foot her expenses, the privileged, outspoken, headstrong, and educated Aisha has the freedom and the gumption to make financial decisions, however small they may be. In finally taking her to the brink of economic independence at the close of the film, *Aisha* marks a great twenty-first-century leap. Aisha's qualities are just awaiting optimum application in the correct sphere. Marking this development, the closing voiceover in Aisha's voice is very different from the initial impish voiceover, delivered with her face turned directly towards the camera, looking straight at the audience. Whereas in the first one, she talks of her ongoing flirtation with the metatextual Austen–*Emma* pursuit of matchmaking—'These

days this wedding business has become so *painful*, isn't it?... the real *satisfaction* is when you get two people together. I mean, to *perfectly match two perfect people*', in the end, she is 'older and one year wiser' and has 'understood that love doesn't come with planning'. This makes Aisha something more than the 'heroine of surfaces and mirrors' that Theresa Kinney accuses her of being, with that realization, very close to Austen's Emma.[10]

*Aisha* then marks an updated convergence of cultures implied in the give and take between *Emma*, *Clueless*, and *Aisha*. This mainstream Bollywood offering brings the best of the Bollywood format to the core of the two previous outings. The nuancing that takes place between the word and the image, between the image and the other images, in the transposition from one physical and notional cultural space, gives rise to enriching the interstitial complexities. The multifaceted agenda of the adaptive engagement, underlying this pulsating parley between multiple disparate cultures and existing between two mediums, is fascinating enough, but it is supplemented by the fraught yet relevant dynamic of a hybrid cosmopolitanism, which places *Aisha* at a thought-provoking, nuanced intersection. The complexities arising from the matrix of this vibrant interface and from the underlying agenda of the adaptive engagement, enables the Indian filmic adaptation of *Emma* to successfully move out of the ambit of the restrictive anxiety of influence discourse, even as it transforms, indigenises, and enriches the text interpretationally, becoming an independent witness to the immense economic and sociological changes and developments that have come with the new millennium in the nation. In sync with Sinyard's belief that 'Adapting a literary text for the screen is essentially an act of literary criticism' (Sinyard 1986), *Aisha*, very cleverly, turns both the creative and the critical spheres of adaptation to its advantage, a space that has been, over the years, both much denigrated and much applauded. If *Aisha*, even while looking back to both its literary and filmic predecessors, has yet managed to carve its own space, it has achieved something singular. Opening out 'a sensorium of adaptational experience',[11] it not only carries Austen and Heckerling along, it also manages to creatively and critically distance itself from them. It is not merely reiterative imitation nor merely mindless plagiarism, but an entirely new entity, successfully countering the 'fidelity discourse' about adaptations and overturning the dismissal of the adaptive exercise as a redundant, reductive, and 'secondary' one. Far from murdering the original, to use Hindi film director Chandraprakash Dwivedi's grouse against adaptations, *Aisha* is 'second' (actually, third), without being 'secondary'.[12] It actually ends up re-invigorating Austen, ever 'respectful' to the original in line with Young's proposition,[13] by giving the literary original extended life, marking its continued relevance in a completely new cultural scenario.[14] No longer playing 'second fiddle' to either the literary original or its filmic predecessor, this 'deliberate'[15] Indian adaptation of Austen's *Emma* stands on its own, offering its own unique insights and becoming a contemporary classic in its own right.

## 5. Postscript

Two years after the release of *Aisha*, Rajshri Ojha herself had washed herr hands of the film, saying '*Aisha* was not my film'. In addition, that 'the film that I had conceptualized was not the one the audience saw'.[16] Moreover, 'When we decided to do an adaptation of Jane Austen's *Emma*, there were many layers to the character. However, I don't know where those layers vanished'. To me, since *Aisha* manages to accomplish so much, in assuming an identity beyond that intended by its maker, this indicates that not only does this version of *Emma* legitimate itself, but there is enough ground for another to give voice to the 'layers' that could not come through in this one.

**Funding:** This research received no external funding.

**Institutional Review Board Statement:** Not applicable.

**Informed Consent Statement:** Not applicable.

**Data Availability Statement:** Not applicable.

**Conflicts of Interest:** The authors declare no conflict of interest.

## Notes

1 This connection has been noted by Douglas Lanier in connection with adaptations of Shakespeare in (Lanier 2014).

2 The edition used for this essay is the Chapman (2015).

3 Hutcheon, *Adaptation*, 8.

4 The six films with the actors playing Emma or her character adaptation are: 1948 (Judy Campbell); 1995 (Alicia Silverstone in *Clueless*); 1996 (Gwyneth Paltrow); 1996 (Kate Beckinsale in a TV film version); and 2010 (Sonam Kapoor in the Bollywood adaptation, *Aisha*); 2020 (Anya Taylor-Joy). TV series: 1960 (Jane Fairfax); 1972 (Doran Godwin); 1996–1999 (TV adaptation of *Clueless*, Rachel Blanchard); and 2009 (Ramola Garai).

5 *The Cambridge Introduction to Jane Austen* has a chapter by Janet Todd entitled 'Austenmania: Jane Austen's global life', pp. 142–51. The use of this term was consolidated by Anna Leszkiewicz when they talked of the spurt in Austen adaptations in 1995 in an article for (Leszkiewicz 2015). Leszkiewicz further notes how 'Journalists were stunned, variously labeling the rush of popularity 'Austenmania', 'Austenfever', 'Austenitis', and 'Darcymania'. The spurt in the popularity of Jane Austen has led to a dedicated fan-following, called 'Janeites'.The word has even found itself into the Urban Dictionary, where it has been defined as referring to 'the fact that these books, especially *Pride and Prejudice*, have become the model for most romance novels today and that we still make movies and remakes of it. Despite most of the public being ignorant of the existence of Ms. Austen, our culture is riddled with her influence' (https://www.urbandictionary.com/define.php?term=Austenmania, accessed 6 January 2022). This definition goes on to assert: 'We will never, ever get rid of Austenmania. The Austen is ingrained in our souls'.

6 Sharmistha Ghosal, Interview, 'Filmmaker Rajshree Ojha is ready to make a film adapted from Jane Austen's Sense and sensibility', *Cinema Express* (13 November 2021). https://www.cinemaexpress.com/international/interviews/2021/nov/13/filmmaker-rajshree-ojha-is-ready-to-make-a-film-adapted-from-jane-austens-sense-and-sensibility-27825.html (accessed on 18 May 2022)

7 Maasi, literally 'like mother', refers to the mother's sister. Her husband becomes the 'maasu'. In Indian languages, there are specific names for all relationships that become open announcements of them.

8 Poo, Pooja in Karan Johar's 2001 blockbuster, *Kabhi Khushi Kabhi Gham*, was the girl with an attitude played by Bollywood star, Kareena Kapoor. This character went on reach iconic status in the annals of Bollywood history.

9 Kenney, 'Not entirely Clueless'.

10 Kenney, 'Not Entirely Clueless'.

11 Leonore Lieblin, review of Linda Hutcheon's *A Theory of Adaptation: Borrowers and Lenders, The Journal of of Shakespeare and Appropriation* III: 1 (Fall/Winter 2007) (www.borrowers.uga.edu/781621/show, accessed 30 June 2019).

12 Chandraprakash Dwivedi, successful director of *Pinjar* (Dwivedi 2003), an adaptation of a 1950 Punjabi novel on the partition by Amrita Pritam, had made an early comment on what it felt like watching a filmic adaptation in an interview shortly after the release of the film: 'Whenever a film is on a literary subject, it is like watching literature being murdered'. However, the director has gone on to adapt various works on literature to prove the lie to this erroneous belief. They strongly believe: 'Literature is rich with well-conceived ideas and plots that are elaborately thought out. Therefore, adapting a novel readily ensures strong content' (Khanna 2005).

13 James O Young, op. cit., p. 144.

14 Hutcheon, *Adaptation*, p. 9.

15 Hutcheon, *Adaptation*, xiv.

16 '*Aisha* was not my film', interview in the *Mumbai Mirror*, Entertainment Times, *Times of India*, 9 March 2012. https://timesofindia.indiatimes.com/entertainment/hindi/bollywood/news/aisha-was-not-my-film/articleshow/12194625.cms#:~:text=Rajshree%20Ojha%2C%20who%20directed%20Anil,the%20theatres%20wasn\T1\textquoteright t%20hers.&text=Nevertheless%20Ojha%20is%20bouncing%20back,eminent%20Hindi%20litterateur%20Nirmal%20Varma (accessed on 18 May 2022).

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
