# Peer review of "Did We Need Another Emma? The Anxiety of Influence in the Bollywood Adaptation of Emma"

_humanities, doi:10.3390/h11040080_

Round 1

Reviewer 1 Report

This article is a fine piece of research. The author is aware of previous scholarship on the film Aisha, and uses it well throughout the discussion. The hypothesis of the article is good and so is the discussion.

That said, there are certain aspects that need to be tackled before accepting this essay.

First of all, sections should be included. I think the article would certainly benefit from having them.

The introduction needs to be improved. As it is now, the reader does not really get a good idea of what the essay is going to be about and the introduction does not really catch the reader's attention. However, the discussion itself is extremely interesting. 

When talking about the interconnections among Aisha, Clueless and Emma, I was all the time thinking about the idea of the rhizome by Deleuze and Guattari. I strongly recommend that the author reads "Shakespearean Rhizomatics: Adaptation, Ethics, Value" by Douglas Lanier, since it can be extremely useful for this article. Although Lanier applies the idea of the rhizome to Shakespeare, the author can apply this theory to this essay.

I also think the author should revise the bibliography. In the works cited section, there are many articles that have not been cited throughout the essay, and they should.

As for the Use of language, the article is well-written in general and quite interesting. However, the author should revise the use of punctuation (see, for instance, page 6, 4th paragraph). There is also a mistake on page 4, at the end of the first paragraph since the word quoting has been repeated twice.

Anyway, all in all, this is a good paper and provides fresh material to Austen studies. 

Author Response

Section headings have been introduced as suggested.

The Introduction has been made pointed to state the interest in the premise. 

Deleuze and Guattari's 'rhizome' idea proposed in A Thousand Plateaus: Capitalism and Schizophrenia, and taken up by Douglas Lanier for Shakesperean adaptations, has been tapped. 

The bibliography has been changed to fit in with 'Works Cited' rubric.

Reviewer 2 Report

This article discusses the 2010 film Aisha as the product of layers of adaptation of Jane Austen’s Emma.  The essay is provocative, and I offer two suggestions for revision.

The argument that Aisha is both influenced by and moves beyond the novel and the 1996 adaptation Clueless is well-made but not fully articulated until the end of the essay (page 9): “Aisha turns both the creative and the critical spheres of adaptation to its advantage, a space that has been, over the years, both much denigrated and much applauded. If Aisha, even while looking back to both its literary and filmic predecessors, has yet managed to carve its own space, it has achieved something singular. Opening out ‘a sensorium of adaptational experience’ it not only carries

Austen and Heckerling along, it also manages to creatively and critically distance itself

from them.” Presenting these points earlier will help the reader navigate the essay and understand where the subsequent analysis is headed.

The essay provides interesting context for Austen adaptations in Indian cinema, and additional examples and discussion of the popular and critical reception of Aisha on page 2/footnote 7 would be helpful as the essay seems to be writing back against some of these responses.

Author Response

The suggestion of stating the premise directly, right at the outset, is well taken and followed.

Footnote on the reception of Aisha augmented

Reviewer 3 Report

I think there is a compelling argument to be gleaned from this. In my rendering, it goes something like this: "Aisha takes Emma's concern with shifting (but finally conservative) class relations and, like Clueless, has them play out cinematically among characters in contemporary, urban settings. While Clueless relocates Austen's drama from Regency England to the US in the 1990s, Aisha takes place in twenty-first century India. This essay describes the various class markers, especially linguistic ones, carefully deployed by the Hindi film--distinctions of region and rank that may not be legible to viewers outside of India. While drawing on Austen and, especially, Heckerling, Aisha's originality lies in its depiction of the Delhi-Indian urban elite at whom the movie is primarily aimed." Unfortunately, while the author's discussion of class markers is very observant and persuasive, the larger argument outlined above is murky at best. The scattershot comments on adaptation need firmly to reject the criterion of "fidelity" before stating forthrightly how the precedent of Clueless does and doesn't apply. Both Clueless and Aisha need to be introduced to readers who can't be assumed to know basic information about these two texts. (This would help to explain, too, why the far more "faithful" 1996 Emma isn't discussed at all.) Similarly, Aisha's characters need to be identified on first mention with their Austenian counterpart; after all, part of the essay's project is one of translation. If the essay struggled in these ways to frame its argument, it also waffled in drawing conclusions about the movie's class/gender politics. Is Aisha's class "bubble" criticized or does the film (like the novel) reassert old hierarchies by realigning Shefali-Harriet with her own "type"? Does it support Aisha's financial independence or reinforce Arjun-Knightley's superior judgment in keeping traditional "family values"? If the film, like the original, is contradictory on these matters, then that conclusion should be spelled out. In general, I'd encourage the author to begin by asserting their sound intention to defend the film (subordinating claims about adaptation) and then proceed to do so a bit more consistently and systematically. In the service of clarity, I might also suggest streamlining prose whenever possible.

Author Response

re-reading my essay, I find that class distinctions and rank are quite clear even to 'outsiders.' 

The 'fidelity' discourse has been taken up critically in a balanced manner.

The original equivalent characters inserted right at the outset though they became clearer later in the essay as it developed.

Round 2

Reviewer 3 Report

My primary concern regarding the first version of this essay had to do with clarity of argument, the need for a stronger introduction and more sustained through-line. This version does a good job of addressing that concern. The new opening, clearly identifying the three texts that will be in play and the major issues to be addressed, now orients the reader upfront. The addition of apt subheadings is enormously helpful in structuring and signposting the essay as a whole. As indicated before, the most cogent and original part of the essay explains Aisha's various strategies for transposing Austen's class differences to a contemporary Indian context. This section ("Living in a 'Bubble') is now that much more effective for being more strongly framed. Finally, the author's positioning of themselves in relation to debates around "fidelity" feels more confident in this revised version.